# Water Productivity of Phoenix Metropolitan Area Cities

Benjamin L. Ruddell[1], Richard Rushforth[1], Diane Hope[1]

[1] School of Informatics, Computing and Cyber Systems, Northern Arizona University, Flagstaff, Arizona, USA.

*Correspondence to*: Benjamin L. Ruddell (Benjamin.Ruddell@nau.edu)

**Abstract.** Water productivity (or efficiency) data informs water policy, zoning and planning along with water allocation decisions under water scarcity pressure. This paper demonstrates that different water productivity metrics lead to different conclusions about who is using water more effectively. In addition to supporting the population's drinking and sanitation needs, water generates many other public and private social, environmental, and economic values. For the group of municipalities comprising the Phoenix Metropolitan Area we compare several water productivity metrics by calculating the Water Value Intensity (WVI) of potable water delivered by the municipality to its residential and non-residential customers. Core cities with more industrial water uses are less productive by the conventional efficiency measure of water used per capita, but core cities generate more tax revenues, business revenues, and payroll revenues per unit of water delivered, achieving a higher water productivity by these measures. We argue that policymakers should consider a more diverse set of socio-economic water productivity measures to ensure that a broader set of values are represented in water allocation policies.

**Graphical Abstract**

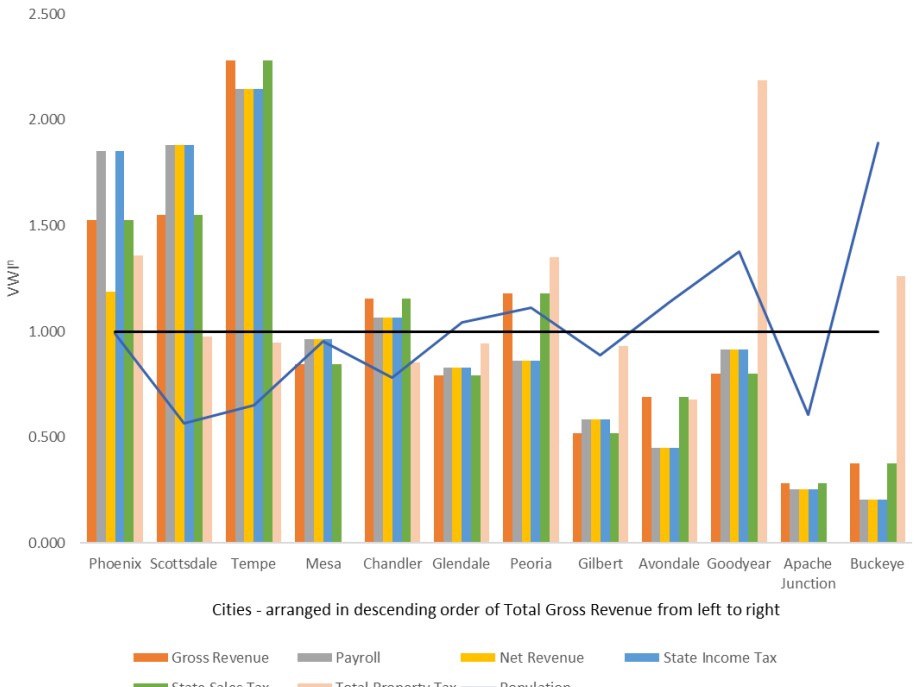



## 1 Introduction

The coming decades will see major challenges in meeting demands for water in the United States and across the globe (Postel, 1996; Devineni et al., 2015). Apportioning water effectively between agriculture, the world's largest water user and the water demands of industry, energy and urban development will become increasingly important (Hoekstra, 2014; Vörösmarty, 2000; Gleick and Palaniappan, 2010). Reliable metrics are needed for informed decision-making about allocating water sustainably, equitably, and optimally. This is especially true in water-scarce regions like the American Southwest (Tidwell et al., 2012; Wildman and Forde, 2012; Schewe et al., 2014). However, in such regions, there is often a limit to how much water cities can reduce through conservation measures or other demand management policies - a phenomenon known as 'demand hardening'. Even if conservation is still producing water efficiency gains decoupled from growth to date (Richter et al., 2020), demand will eventually harden, so it is in the public's interest to allocate water based on the merit and benefit of use (Howe and Goemans, 2007), however merit and benefit might be defined.

Careful management of freshwater is especially important for the municipalities comprising the Phoenix Metropolitan Statistical Area (Phoenix MSA or PMA), Arizona (Gober et al., 2010; Gober et al 2013; Rushforth and Ruddell, 2015). With a population of 4.9 million, in 2019 Phoenix-Mesa-Chandler is the 10th most populous metropolitan area in the country (US Census Bureau, 2020). Economic growth has been tightly coupled with population growth in the PMA. In 2017 the GDP for the Phoenix MSA was close to $217 billion, having grown by 30% between 2010 and 2018 (US Bureau of Economic Analysis, 2019). Underlying the Phoenix MSA's population growth and economic growth are increasingly scarce water resources.

Studies of water use often employ variations of water footprint analysis to measure water use or water use efficiency (Hoekstra et al., 2011; Hoekstra et al.; 2015; Marston et al. 2018; Paterson et al., 2015; Rushforth and Ruddell, 2018). Water footprints have been calculated for cities in the US (Paterson et al. 2015), and even specifically for cities in Arizona (Bae and Dall'Erba, 2018; Rushforth and Ruddell 2015, 2016; Scott and Pasqualetti, 2010). Water productivity studies have been conducted on industries and products (Marston et al., 2020; Evenson et al. 2018; Maupin et al. 2014; Mayer et al., 2016; Blackhurst et al. 2010; Solley et al. 1983), on the electric power grid (Ruddell et al. 2014), and on Arizona semiconductors (Hubler et al 2012), in addition to the more common study of irrigation agricultural water productivity (Xu et al., 2019; Kinje et al., 2003; Hamdy et al., 2003). Water efficiency benchmark data can help policy makers to develop and implement sound water policy (Berg, 2010). Such benchmarks can help stakeholders to quantify progress towards policy objectives and can help regulators fine-tune efficiency goals (Haider et al., 2016). Because we manage what we measure, it is important to inform policy using appropriate measures for what we value about water use.

Per the logic of Embedded Resource Accounting (Rushforth et al. 2013; Ruddell et al. 2014), produced values are accounted for differently by different parties because these parties have different worldviews and decision boundaries by which they account for internal and external costs and benefits. For instance, revenue is mostly valued by business owners, payroll is mostly valued by workers (and is a cost to business owners), taxes are mostly valued by the branch of government collecting the specific tax and by the public beneficiaries of this tax revenue (e.g. state income tax to the state, property tax to the municipality), and population is valued by (presumably) all people – but most especially by democratically elected government officials who set water





58   policy because people vote. There are also many other social, environmental, and economic values produced
59   where water inputs are an input factor (Vardon et al., 2012), including for instance aquatic habitat created by
60   outdoor water use in a desert city, urban heat island mitigation, and federal tax revenue. The return of revenue
61   directly to a water department responsible for its provision is another important type of value needed for fiscal
62   planning and support of water operations (Borrego-Marin et al., 2016), but that kind of revenue is of very
63   narrow interest to a single department of a single municipal government and is discounted by other parties.
64   Because there are many social, environmental, and economic stakeholders with many different sets of interests
65   and values, multiple water use efficiency or productivity benchmarks are appropriate to measure the efficacy of
66   water allocation.

67   The standard residential water efficiency or water sustainability measure for water utilities in the United States
68   is Gallons per Capita per Day (GPCD). Water use efficiency is the reciprocal of the water productivity. Water
69   productivity – also called water value intensity (WVI, Ruddell et al. 2014) is a metric expressing the benefits of
70   water use (in units of the benefit) relative to the costs (in units of water use). The goal of water policy should be
71   to do more social, environmental, and economic good with limited water resources, but not necessarily to use
72   less water. Shifting to a water productivity (or WVI) perspective puts the emphasis on the values and benefits
73   that are produced, rather than the water that is saved. For example, if we invert the standard GPCD metric, we
74   obtain People per Gallon per Day (PPGD), and this makes it clear that such a metric values supporting
75   additional population using the water resources. It is not incorrect to use an efficiency metric, but we prefer the
76   clearer productivity framing for these reasons.

77   Comparing multiple water productivity metrics and benchmarks is particularly helpful when there are multiple
78   values and benefits associated with the water use. In this paper we develop a case study comparing multiple
79   water productivity benchmarks for the group of municipalities comprising the Phoenix Metropolitan Area. For
80   these municipalities we compare the water productivity in units of value produced per acre-foot of water
81   delivered. Water productivity metrics in this paper's case study include (1) residential population supported, (2)
82   payroll, (3) net revenue, (4) gross revenue, (5) net revenue, (6) state income tax, (7) state sales tax, and (8) total
83   property tax. Other productivity metrics could be used such as the water intensity of land use, or we could add
84   more social and environmental value considerations, but these are beyond the scope of this paper's case study
85   due primarily to a lack of data availability. Our research question is, "What is the comparative water
86   productivity of the municipalities of the Phoenix area, using multiple water productivity measures?"

## 2 Methods

Water that is available to PMA cities is allocated using a complex system of legal water rights and conveyed to
the municipalities via large-scale physical infrastructure systems (Jacobs & Megdal, 2004; Holway, 2007). Most
PMA municipalities draw water from three main physical water sources: the Colorado River, the Salt-Verde
River system, and the large, interconnected groundwater aquifer underlying the metro area. However, while
many municipalities have access to all three sources, some municipalities, typically newer ones on the outer
edge of the metropolitan area, may not have access to SRP or CAP water (Rushforth et al., 2020).





Within each municipality water is allocated to Residential and Non-Residential uses, which yield residential
values (income tax, property tax, population) and non-residential values (payroll, net/gross revenue, sales tax).
Of the many municipalities comprising metropolitan Phoenix, we include twelve in this study (Figure 1):
Apache Junction, Avondale, Buckeye, Chandler, Gilbert, Glendale, Goodyear, Mesa, Peoria, Phoenix,
Scottsdale, and Tempe. Other smaller and outlying cities (e.g. Litchfield Park, El Mirage, Paradise Valley,
Queen Creek, Guadalupe, Surprise, Cave Creek, Fountain Hills) were omitted due to a lack data.

Water use studies may be based on consumption or withdrawal accounting. This study uses withdrawal
accounting, and specifically water delivered to utility customers, rather than net consumptive use. This is the
right choice for most water use studies per the arguments in Ruddell (2018), because city water resources,
infrastructures, operating costs, and water rights are measured and priced in units of water volumes withdrawn
and delivered, not in terms of net hydrological water balances. We use acre-feet units for this study, not SI units,
because acre-feet is the unit of measurement used and understood throughout the water management community
in the USA and converting to SI units renders the results more difficult for use in policy applications. Reclaimed
water use was not included in this study since it is not allocated to municipalities by an external agency (e.g.
SRP, CAP, or ADWR in this case), and because it is not withdrawn from the three major hydrological water
sources of the region. Also, reclaimed water generally is used low economic value or indirect economic value
activities such as recreational turf irrigation, making it difficult to measure associated economic value. And,
because reclaimed water use (unlike raw water deliveries) is subject to inconsistent city and county policies for
reporting and accounting, it is difficult to compare reclaimed water data robustly between municipalities.

The water productivity metrics relate value output to water input using six different Water Value Intensities
(WVIs): the residential population supported, along with six different financial metrics: net revenues; gross
revenues, payroll, net revenue, state sales tax, and state income tax, and property taxes. Value intensities could be
calculated various ways, using a range of different or additional metrics – to include for example, different
social and environmental benefits of a city's water use. Also, the metrics could potentially be weighted to assign
the differential importance to some values versus others. For the purposes of this study, we used metrics that
were readily available and reliable (Table 1). Payroll and taxes are components of gross revenue. We present
both gross revenue and net revenue reported by the economic census.

**2.1 Data Sources**

This study uses older data from calendar year 2007 due to data availability constraints. The specific variety of
data for residential and non-residential water use was no longer collected by the State of Arizona after 2009. We
chose 2007 because this is the most recent pre-2009 year coinciding with the publication of the U.S. Census
Economic Census.

Residential and non-residential water use data for the PMA's municipalities in this study were obtained from the
Arizona Department of Water Resources Imaged Records. Reported water use data for 2007 were used to match
US Economic Census data for the same year. Specifically, water use data contained in this report is found in
ADWR Notifications on Gallons Per Capita per Day (GPCD) and Lost and Unaccounted (L&U) for Water sent
to the individual cities studied in this report (ADWR, 2011a-i). L&U water was incorporated into this study by



attributing L&U water proportionately to total water use by residential and non-residential sectors (for an

example see Appendix A-1 and the equation in Appendix B).

Income data were obtained from the U.S. Census Bureau (2009a-f). Property tax data were obtained from the

annual budgets from each of the cities in the study (City of Chandler, 2008, 2009; City of Glendale, 2008; City

of Goodyear, 2007; City of Mesa, 2008; City of Peoria, 2007; City of Phoenix, 2007; City of Scottsdale, 2008;

City of Tempe, 2007; Town of Avondale, 2010; Town of Buckeye, 2007; Town of Gilbert, 2007).

Manufacturing, retail, information services, real estate, and professional and technical services data were

obtained from the 2007 Economic Census (U.S. Census Bureau, 2009a-f). See Appendix C for the full economic

data used in this study.

Water Value Intensities (WVIs) were calculated using the water-volume weighted averages of residential and

non-residential sectors (Table D1). Economic values on a water use basis were analyzed for six economic

categories in the U.S. Economic Census: city-level or town-level income data (Tables D2, D3), city- or town-

level manufacturing (Tables D4, D5), city- or town-level retail data (Table D6), city- or town-level information

services (Table D7), city- or town-level real estate data (Table D8) and city- or town-level professional and

technical services (Table D9).

**2.2 Simplified Embedded Resource Accounting: or, Point of View Matters in Water Use Accounting**

This analysis employs a simplified version of Embedded Resource Accounting (ERA, Ruddell et al. 2014) to

associate indirect and direct values with direct and indirect impacts in an input-output network. In this case there

are six direct and local values produced, one direct impact on the local freshwater stock, and indirect values and

impacts are neglected. The water use metrics in this paper are therefore calculated from the point of view of a

hypothetical manager of the water resources of the Phoenix Metropolitan Area who is interested in maximizing

a diverse basket of values that are directly associated with water use processes in the PMA. The same

hypothetical manager is therefore also disinterested in in indirect value creation and impact such as federal tax

revenues or the water impacts of the PMA's supply chains lying outside the area. Everything inside the PMA is

"internal" and everything outside the PMA is "external" from this hypothetical manager's point of view. We

assert that this point-of-view is historically responsible for water allocation decisions and regulations for the

PMA, and resembles the point of view of the Governor's office, the regional government, or the Arizona

Department of Water Resources, so this is an appropriate choice for this study. Because the worldview of this

hypothetical manager encompasses the metro area, ERA defines the resource stock of interest as the total

combined annual water deliveries from the Central Arizona Project (2012), Salt River Project (SRP), and

groundwater resources to the PMA's major municipalities individually and collectively. If a different point of

view is chosen for the accounting, the results will change. For example, the business owners of the City of

Tempe internalize revenue-generating value, but not necessarily other values like payroll or taxes benefitting the

City of Tempe and its labor force.

The direct water value intensity $WVI_{x,l}$ used here is simply the ratio of the value ($V$) of type ($l$) produced as an

output of the municipality's ($x$) collective processes to the input of water ($W$) to the municipality's processes. In

other words, $WVI_{x,l}$ is the ratio of value out to water in. $\overline{WVI_l}$ is the mean WVI for value $l$ for all municipalities





in the area. $WVI^n_{x,l}$ has been normalized ($n$) by dividing $WVI_{x,l}$ by the mean $\overline{WVI_l}$, such that municipalities

with results above 1 have above-average WVI for that value type. $BWVI_x$ is the basket-weighted water value

intensity for municipality $x$; it is the weighted average across all value types for that municipality. In this study,

we assume weights of 1 for all value types. From this point of view, all six types of value assessed here are

weighted equally. $BWVI_x{}^n$ is the normalized value, like $WVI^n_{x,l}$ above.

*WVI*'s may include economic data and measures of economic value, but a *WVI* – or any *VI* – is not a price or a

measure of marginal value or cost according to the classical economic *Theory of Value*, because it does not

consider the marginal contribution of the impact on the resource stock to the production of values, or the cost of

the resource, or value-added by the process. Because *VI*'s are not prices or costs, they may not be added together

to directly measure the value produced by a process; rather, a basket (i.e. a range) of *VI*'s should be interpreted

as multiple independent objectives, e.g. using a Pareto framework, or assigned relative weights by a decision

maker.

### 2.2 Residential Sector Water Value Intensities

Property taxes were used as a measure for the values produced by residential water use. Primary, secondary, and

total levied property taxes by municipalities were considered in this analysis. Calculation of the value intensity

of residential water on a per volume use basis is shown in Appendix A.

### 2.3 Non-Residential Sector Water Value Intensities

City-level net and gross revenues and payrolls were used as a measure for the values produced by non-

residential water uses such as commercial, industrial, and governmental uses of the city's potable water

supplies. City-level state sales tax contributions and income taxes paid to the state were estimated for the non-

residential sector using the gross revenue and payroll data, respectively. The state sales tax rate was set at 6.6%

and the income tax rate 3.3%, per statutes in effect in Arizona during the study period. From these data, the

value intensity of non-residential water uses was calculated for city-level net/gross revenues, payroll, state sales

tax contribution, and income taxes paid to the state. Note that income tax is considered a value product of the

non-residential sector in this analysis, and taxed payroll is a value product of the business sector, not the

residential sector. Net and gross revenue and payroll data were obtained from the US Economic Census.

Population data were obtained from the U.S. Census Bureau (2007b). Equations for Revenue, Payroll, and Tax

VI's follow. Calculation methods are show in Appendix A.



**3 Results**

In terms of residential population supported per acre-foot of water used (Figure 2), outlying cities such as Buckeye, Goodyear and Avondale are more productive (or efficient) than core cities like Phoenix, Tempe and Scottsdale. However, when economic productivity measures are considered (Figure 3), core cities like Phoenix, Tempe and Scottsdale, dominate the rankings because they produce far more payroll, tax, and business revenue per gallon of water used.

**4 Discussion**

Each city has its own unique water value profile (Table 1) which contribute to its water productivity profile. For example, Chandler is the fourth largest city in the PMA by population, and had the fourth lowest WVI per capita, but its WVI for gross revenue is well above the PMA average (Figure 3). Chandler has a disproportionately large industrial sector dominated by High Value Semiconductor Manufacturing products and services. Previous studies have found this sector produces an unusually large amount of economic value relative to use of water (Hubler et al., 2012). Figure 3 reveals tradeoffs between multiple water productivity objectives. For example, there is a tradeoff between WVI for net revenue versus WVI for population. The relatively higher business revenue a community generates with its water, the relatively lower population it supports with its water. A detailed study of the Pareto frontiers and tradeoffs between these multiple objectives is beyond the scope of this paper, but such a tradeoff appears to have emerged within the PMA. Despite this, the standard U.S. measure of water efficiency, Gallons per Capita per Day, (GPCD, Evenson et al., 2018), implies that water's value lies entirely in supporting residents and their swimming pools and lawns. When applied in isolation from other metrics for other objectives, this standard measure favors allocating water to bedroom communities. But this comes at a cost of the jobs and tax revenues that the residents of those bedroom communities need for their livelihoods and to pay for their water rights and water infrastructure.

Because cities, state government, and economic development organizations want to promote high-quality economic development, and the City of Chandler uses much of its water for this kind of economic activity, allocating more water toward Chandler as compared with a bedroom community would seem to merit consideration based on economic water productivity benchmarks. After all, a bedroom community's residents need the payroll and tax revenues produced by companies in the City of Chandler. But, in turn, those companies employ the workforce that lives in the bedroom communities and depend on that labor for their operations. A residential population cannot be supported without jobs and revenues; both values matter and each supports the other. Therefore, a more diverse set of water productivity benchmarks can help decision makers understand the trade-offs involved in their allocation of water to different kinds of cities, and can help policymakers avoid undervaluing the economic allocations of water that are needed to support employment for the residential population. Additionally, the tax base is the major constraint on the ability of a city to finance water rights and water infrastructure to provide adequate water for its residential population. Linking economic and population growth is important. There have been several advocates for the concept of 'wet growth' (Arnold, 2005) and water-conscious land-use planning (Bates, 2012). Water-conscious economic planning and growth can help to



promote, protect, and restore water sources, and can prevent growth beyond the limits of water resources (Gober
et al. 2010; Larson et al, 2013; Li et al., 2016).

Accurate estimation of the water resources required to "build out" the municipality's zoning and master plan is
crucial part of this land use planning process (Gober et al., 2010; Gober et al., 2013; Larson et al, 2013; Li et al.,
2016). Once land is allocated to a use, the water and land associated with that use cannot be reallocated easily or
inexpensively, if at all (Marston and Cai, 2016). In addition, as a municipality continues to grow, it typically
approaches the "build-out" stage where further changes become prohibitive due to the scarcity and depletion of
land and water resources. Balancing various water productivity values is therefore important in the land use
planning process before development occurs.

We present results that focus narrowly on economic water productivity in the PMA as an alternative to GPCD as
an efficiency metric, but it is preferable to also include broader economic, environmental, and social dimensions
of water productivity. For example, urban tree and shade programs, which are water consumers, may not have
high economic water productivity or generate tax revenue, but they do produce demonstrable ecological service
benefits such as shade, mitigation of air pollution, flood amelioration, and reduced urban heat island effects.
Water planners and decision-makers do not apply equal weighting to their multiple values, so any stakeholder
would have their own weights to apply to the multiple-objective decision process that is implied by the use of
multiple water productivity metrics.

When broader values like revenue, payroll, and tax benefits are factored into water allocation decisions,
different water allocation decisions could emerge. These are political and value-based decisions, not engineering
decisions, but such decisions should be more broadly informed with a broader set of water productivity
benchmarks.

**5 Conclusions**

This study finds that bedroom cities show higher water productivity based on the standard efficiency benchmark
of gallons per capita, but core cities which host large businesses show higher water productivity using a basket
of economic values like taxes, payroll, and business revenues. There may be a tradeoff between these competing
values produced by water use, so a broader basket of water productivity benchmarks could therefore inform
more balanced and equitable water allocation decisions by policymakers.

**Appendices**

*Appendix A: Detailed VI Equations*
Calculation of the VI of residential water as measured by property taxes, on a per volume use basis $VI_{Property\ Tax}$
was taken by dividing the amount of levied property taxes by the municipality's volume of water allocated to
residential uses, property tax data in Appendix C, were obtained from the Maricopa County Department of
Finance (2007); property taxes reported as zero are due to city-specific policies that restrict the ability of the city
to collect property tax.



$$VI_{Property\ Tax} = \frac{\$\ Levied\ Property\ Tax}{Volume_{H_2O}Residential_i\ (ac-ft)}$$

Per capita water use by the residential water use sector of a municipality $VI_{Population}$ is calculated as shown in

Equation 13. This metric is included because per-capita equity in water use is currently the primary type of

value intensity utilized for water allocation decisions.

$$VI_{Population} = \frac{Population}{Volume_{H_2O},Residential_i\ (ac-ft)}$$

Calculation of the VIs for net and gross revenues, payroll, sales tax and income taxes using the data shown in

Appendix C:

$$VI_{Revenues} = \frac{\$Revenues}{Volume_{H_2O}Non-Residential_i\ (ac-ft)}$$

$$VI_{Payroll} = \frac{\$\ Payroll}{Volume_{H_2O}Non-Residential_i\ (ac-ft)}$$

$$VI_{Sales\ Tax} = \frac{\$\ Gross\ Revenues_i \times State\ Sales\ Tax\ Rate}{Volume_{H_2O}Non-Residential_i\ (ac-ft)}$$

$$VI_{Income\ Tax} = \frac{\$\ Payroll_i \times State\ Income\ Tax\ Rate}{Volume_{H_2O}Non-Residential_i\ (ac-ft)}$$

*Appendices B, C, and D: Source Data Tables*

Appendix B: Water Data Tables B1-B3

Appendix C: Tax Data Tables C1

Appendix D: Financial Data Tables D1-D9





Table B.1. Reported Total Water Demand for PMA Municipalities Included in this Study.

| City | Demand Category | Year | | | | | | | | | |
| --- | --- | --- | --- | --- | --- | --- | --- | --- | --- | --- | --- |
| | | 2000 | 2001 | 2002 | 2003 | 2004 | 2005 | 2006 | 2007 | 2008 | 2009 |
| Apache Junction | Total | 10,627 | 10,523 | 11,416 | 10,983 | 10,639 | 11,396 | 11,251 | 11,825 | 11,112 | 11,144 |
| Avondale | Total | 5,653 | 7,758 | 9,295 | 10,040 | 11,123 | 9,893 | 13,378 | 14,185 | 13,033 | 13,277 |
| Buckeye | Total | 1,094 | 1,049 | 2,434 | 2,601 | 738 | 751 | 3,028 | 4,135 | 4,363 | 4,277 |
| Chandler | Total | 48,969 | 53,263 | 55,475 | 55,657 | 55,697 | 58,439 | 61,070 | 64,404 | 63,076 | 60,773 |
| Gilbert | Total | 30,438 | 32,800 | 33,984 | 38,047 | 36,596 | 40,190 | 50,515 | 47,915 | 49,085 | 46,239 |
| Glendale | Total | 49,472 | 49,773 | 51,193 | 48,707 | 48,828 | 49,242 | 49,740 | 46,849 | 49,586 | 48,133 |
| Goodyear | Total | 2,570 | 3,309 | 3,555 | 4,243 | 5,307 | 6,328 | 6,409 | 8,088 | 8,163 | 8,289 |
| Mesa | Total | 101,461 | 102,935 | 97,180 | 100,458 | 95,933 | 100,363 | 100,203 | 100,027 | 93,317 | 89,794 |
| Peoria | Total | 24,602 | 21,503 | 22,593 | 21,715 | 22,656 | 25,421 | 27,659 | 28,527 | 28,717 | 27,388 |
| Phoenix | Total | 332,038 | 340,870 | 346,226 | 329,939 | 337,412 | 314,314 | 331,174 | 321,476 | 304,153 | 305,124 |
| Scottsdale | Total | 79,479 | 78,165 | 84,508 | 77,901 | 74,426 | 80,772 | 84,427 | 85,249 | 84,051 | 83,444 |
| Tempe | Total | 63,236 | 61,729 | 60,223 | 58,526 | 57,644 | 53,515 | 52,201 | 54,915 | 50,239 | 49,682 |





Table B.2. Reported Residential Water Demand for PMA Municipalities Included in this Study.

| City | Demand Category | Year | | | | | | | | | |
| --- | --- | --- | --- | --- | --- | --- | --- | --- | --- | --- | --- |
| | | 2000 | 2001 | 2002 | 2003 | 2004 | 2005 | 2006 | 2007 | 2008 | 2009 |
| Apache Junction | Residential | 4,701 | 4,917 | 5,387 | 5,605 | 5,678 | 5,804 | 6,059 | 6,059 | 6,059 | 5,761 |
| Avondale | Residential | 4,835 | 5,481 | 6,119 | 6,483 | 7,175 | 7,093 | 8,362 | 8,362 | 8,832 | 8,715 |
| Buckeye | Residential | 581 | 604 | 679 | 622 | 599 | 643 | 1,617 | 1,617 | 1,617 | 2,629 |
| Chandler | Residential | 27,488 | 29,152 | 31,316 | 31,599 | 32,465 | 33,906 | 35,539 | 36,563 | 34,424 | 34,766 |
| Gilbert | Residential | 19,816 | 21,702 | 23,905 | 24,647 | 25,633 | 27,110 | 28,684 | 28,684 | 28,684 | 30,910 |
| Glendale | Residential | 35,135 | 34,667 | 36,044 | 34,348 | 34,427 | 33,567 | 34,660 | 34,660 | 34,660 | 31,457 |
| Goodyear | Residential | 1,335 | 1,640 | 2,006 | 2,430 | 3,086 | 3,481 | 3,883 | 3,883 | 3,883 | 4,397 |
| Mesa | Residential | 64,242 | 65,180 | 67,026 | 65,655 | 65,890 | 63,972 | 65,319 | 65,139 | 65,139 | 60,494 |
| Peoria | Residential | 14,400 | 15,208 | 17,077 | 16,925 | 16,962 | 16,224 | 18,981 | 18,981 | 18,981 | 18,819 |
| Phoenix | Residential | 208,431 | 205,247 | 209,018 | 201,004 | 200,214 | 195,013 | 202,387 | 202,387 | 202,387 | 188,503 |
| Scottsdale | Residential | 49,659 | 49,370 | 52,737 | 51,083 | 46,873 | 54,719 | 57,401 | 57,401 | 57,401 | 56,568 |
| Tempe | Residential | 29,814 | 30,826 | 31,884 | 27,593 | 27,368 | 25,989 | 26,208 | 26,208 | 26,209 | 25,024 |





Table B.3. Reported Non-Residential Water Demand for PMA Municipalities Included in this Study.

| City | Demand Category | Year | | | | | | | | | |
|---|---|---|---|---|---|---|---|---|---|---|---|
| | | 2000 | 2001 | 2002 | 2003 | 2004 | 2005 | 2006 | 2007 | 2008 | 2009 |
| Apache Junction | Non-Residential | 5,419 | 5,137 | 5,585 | 5,183 | 4,741 | 5,145 | 5,048 | 5,048 | 5,048 | 4,748 |
| Avondale | Non-Residential | 2,305 | 2,150 | 2,866 | 2,821 | 3,118 | 1,983 | 4,097 | 4,097 | 3,846 | 4,060 |
| Buckeye | Non-Residential | 301 | 188 | 848 | 1,788 | 106 | 110 | 1,270 | 1,270 | 1,270 | 1,482 |
| Chandler | Non-Residential | 18,149 | 19,936 | 20,795 | 20,126 | 19,164 | 20,259 | 22,043 | 23,635 | 23,316 | 21,739 |
| Gilbert | Non-Residential | 6,503 | 8,354 | 8,030 | 9,244 | 9,679 | 9,995 | 11,585 | 11,585 | 11,585 | 11,929 |
| Glendale | Non-Residential | 10,595 | 11,521 | 12,351 | 11,311 | 11,013 | 10,797 | 12,965 | 12,965 | 12,965 | 12,135 |
| Goodyear | Non-Residential | 1,156 | 1,668 | 1,486 | 1,730 | 2,199 | 2,959 | 2,756 | 2,756 | 2,756 | 3,442 |
| Mesa | Non-Residential | 27,053 | 36,579 | 29,500 | 29,028 | 29,252 | 26,898 | 29,373 | 29,373 | 29,373 | 27,340 |
| Peoria | Non-Residential | 4,923 | 4,334 | 3,890 | 3,539 | 4,183 | 5,573 | 7,248 | 7,248 | 7,248 | 7,449 |
| Phoenix | Non-Residential | 102,683 | 102,182 | 105,805 | 100,008 | 101,098 | 106,018 | 109,194 | 109,194 | 109,194 | 102,979 |
| Scottsdale | Non-Residential | 18,730 | 20,071 | 18,740 | 16,140 | 25,392 | 21,305 | 23,725 | 23,725 | 23,725 | 21,274 |
| Tempe | Non-Residential | 27,656 | 26,117 | 24,887 | 25,396 | 25,343 | 23,811 | 24,393 | 24,392 | 24,392 | 22,761 |





Table C.1. 2007 Payroll and Gross Revenue for PMA Municipalities Included in this Study

| Economic Characteristics | Tempe | Scottsdale | Phoenix | Peoria | Chandler | Mesa | Goodyear | Glendale | Avondale | Gilbert | Buckeye | Apache Junction |
|---|---|---|---|---|---|---|---|---|---|---|---|---|
| Population | 172,589 | 233,105 | 1,536,632 | 152,795 | 242,522 | 459,742 | 53,654 | 249,455 | 78,043 | 204,904 | 37,678 | 32,901 |
| Payroll ($1000's) | 138,748 | 188,927 | 700,624 | 28,946 | 80,686 | 113,398 | 8,702 | 48,377 | 7,534 | 32,876 | 990 | 3,364 |
| Gross Revenue ($1000's) | 1,658,541 | 1,750,750 | 6,504,679 | 445,974 | 987,115 | 1,121,299 | 85,776 | 521,636 | 129,608 | 330,023 | 20,513 | 42,344 |





Table C.2. 2007 Tax Data for PMA Municipalities Included in this Study

| Taxes Collected | Tempe | Scottsdale | Phoenix | Peoria | Chandler | Mesa | Goodyear | Glendale | Avondale | Gilbert | Buckeye | Apache Junction |
|---|---|---|---|---|---|---|---|---|---|---|---|---|
| State Income Tax Paid ($1000's) | 4,440 | 6,046 | 22,420 | 926 | 2,582 | 3,629 | 278 | 1,548 | 241 | 1,052 | 32 | 108 |
| Primary Property Tax Paid ($1000's) | 10,371 | 21,166 | 103,664 | 3,002 | 8,506 | - | 4,172 | 3,888 | 1,796 | - | 2,839 | - |
| Secondary Property Tax Paid ($1000's) | 21,365 | 29,673 | 163,227 | 20,527 | 25,109 | - | 6,633 | 24,669 | 4,087 | 27,258 | 347 | - |
| State Sales Tax Paid ($1000's) | 109,463,701 | 115,549,495 | 429,308,843 | 29,434,280 | 65,149,612 | 74,005,744 | 5,661,210 | 34,427,999 | 8,554,147 | 21,781,503 | 1,353,845 | 2,794,705 |





Table D.2. Income Data for Municipalities in the Phoenix Active Management Area (Sources: US Census Bureau and AZ Department of Revenue)

| City | Population | HOUSEHOLD INCOME $ | Average FAGI $ | Per Capita FAGI $ | Average Tax Liability $ | Tax Liability Per Return $ | State Transaction & Priviledge Tax Distributed to Cities $ (2004) | State Sales Tax Per Capita $ | State Transaction and Priviledge Tax Distributed to Cities $ (2011) |
|---|---|---|---|---|---|---|---|---|---|
| Apache Junction | 32,901 | 46649 | 45569 | 23115 | 1039 | 804 | 2661210 | 81 | 2618154 |
| Avondale | 78,043 | 69069 | 45162 | 18332 | 952 | 769 | 3001578 | 38 | 5351475 |
| Buckeye | 37,678 | 71177 | 48216 | 18785 | 1010 | 816 | 710766 | 19 | 2112351 |
| Chandler | 242,522 | 86333 | 63336 | 29647 | 1670 | 1393 | 14770829 | 61 | 17695102 |
| Gilbert | 204,904 | 94151 | 67378 | 28504 | 1703 | 1418 | 9176047 | 45 | 13787266 |
| Glendale | 249,455 | 65769 | 48584 | 21643 | 1159 | 919 | 18303410 | 73 | 1843879 |
| Goodyear | 53,654 | 87264 | 62050 | 26471 | 1388 | 1157 | 1581887 | 29 | 3661678 |
| Mesa | 459,742 | 63739 | 47815 | 21871 | 1141 | 908 | 33254566 | 72 | 34220312 |
| Peoria | 152,795 | 78677 | 58290 | 26943 | 1372 | 1126 | 9064543 | 59 | 10673717 |
| Phoenix | 1,536,632 | 66661 | 50734 | 22341 | 1382 | 1077 | 110504126 | 72 | 112704366 |
| Scottsdale | 233,105 | 106485 | 103539 | 55155 | 3960 | 3100 | 16956076 | 73 | 17843974 |
| Surprise | 87,488 | 68704 | 49775 | 25436 | 1014 | 790 | 2580405 | 29 | 6946254 |
| Tempe | 172,589 | 66359 | 52168 | 28510 | 1401 | 1148 | 13268827 | 77 | 12656738 |
| Tolleson | 6,989 | 41342 | 39084 | 14834 | 807 | 643 | 416070 | 60 | 497422 |



Table D.3. Income Data for Municipalities in the Phoenix Active Management Area (Sources: US Census Bureau and AZ Department of Revenue)

| City | State Sales Tax Per Capita $ | Sales Tax per City Area $ | Sales Tax per GPCD $ | Sales Tax per Ac-Ft of Water $ | Distribution of Income Tax as Urban Revenue Sharing $ | Per Capita Urban Revenue Sharing $ | Urban Revenue Sharing per GPCD $ | Urban Revenue Sharing per City Area $ | Urban Revenue Sharing per Ac-Ft of Water $ |
|---|---|---|---|---|---|---|---|---|---|
| Apache Junction | 79.58 | 74826 | 10287 | 255.58 | 3316127 | 100.79 | 13030 | 94774 | 323.72 |
| Avondale | 68.57 | 117357 | 38703 | 441.58 | 6750611 | 86.50 | 48822 | 148040 | 557.03 |
| Buckeye | 56.06 | 5629 | 15440 | 364.87 | 2427836 | 64.44 | 17745 | 6470 | 419.36 |
| Chandler | 72.96 | 274726 | 108412 | 398.03 | 22468783 | 92.65 | 137659 | 348840 | 505.41 |
| Gilbert | 67.29 | 214055 | 71564 | 310.98 | 17280849 | 84.34 | 89697 | 268295 | 389.78 |
| Glendale | 7.39 | 30742 | 7448 | 26.58 | 23590446 | 94.57 | 95287 | 393305 | 340.12 |
| Goodyear | 68.25 | 19123 | 19754 | 327.83 | 4498039 | 83.83 | 24266 | 23491 | 402.71 |
| Mesa | 74.43 | 250790 | 167542 | 324.49 | 43614424 | 94.87 | 213536 | 319637 | 413.57 |
| Peoria | 69.86 | 61203 | 35609 | 207.51 | 13445840 | 88.00 | 44857 | 77098 | 261.40 |
| Phoenix | 73.35 | 218123 | 515447 | 298.68 | 143647008 | 93.48 | 656962 | 278009 | 380.68 |
| Scottsdale | 76.55 | 97020 | 42832 | 163.61 | 22849062 | 98.02 | 54846 | 124234 | 209.50 |
| Surprise | 79.40 | 65686 | 92569 | 942.12 | 8591077 | 98.20 | 114488 | 81239 | 1165.21 |
| Tempe | 73.33 | 316973 | 34153 | 176.20 | 16137384 | 93.50 | 43545 | 404142 | 224.66 |
| Tolleson | 71.17 | 86508 | 5107 | 650.69 | 632468 | 90.49 | 6494 | 109994 | 827.35 |





Table D.4. Manufacturing Data for Municipalities in the Phoenix Active Management Area (Source: US Census Bureau)

| City | Population | Manufacturers shipments, 2007 ($1000) | Merchant wholesaler sales, 2007 ($1000) | Retail sales, 2007 ($1000) | Retail Sales Per Capita (2007) $ | Accommodation and food services sales, 2007 ($1000) | Number of establish- ments | Sales ($1,000) |
|---|---|---|---|---|---|---|---|---|
| Apache Junction | 32,901 | NA | 24707 | 447477 | 13756 | 36282 | 17 | 24707 |
| Avondale | 78,043 | 0 | 73438 | 1601272 | 20243 | 94656 | NA | NA |
| Buckeye | 37,678 | NA | NA | 215169 | 5676 | 17210 | 9 | NA |
| Chandler | 242,522 | 3956031 | 4585919 | 3608290 | 14787 | 500934 | 220 | 4585919 |
| Gilbert | 204,904 | 415891 | 649322 | 2079066 | 10063 | 191244 | 147 | 649322 |
| Glendale | 249,455 | 912989 | 1013545 | 3627782 | 14457 | 340736 | 135 | 1013545 |
| Goodyear | 53,654 | 185496 | NA | 631710 | 11669 | 105052 | 17 | NA |
| Mesa | 459,742 | 3072462 | 2037336 | 6294523 | 13669 | 753178 | 301 | 2037336 |
| Peoria | 152,795 | 267830 | 251210 | 2340433 | 15135 | 258496 | 56 | 251210 |
| Phoenix | 1,536,632 | 16926892 | 23670515 | 21859505 | 14209 | 3644383 | 1946 | 23670515 |
| Scottsdale | 233,105 | 4806562 | 3445500 | 6645363 | 28447 | 1314297 | 538 | 3445500 |
| Surprise | 87,488 | NA | 20359 | 888224 | 9878 | 115082 | 23 | 20359 |
| Tempe | 172,589 | 5877588 | 7286114 | 6172475 | 35768 | 606835 | 511 | 7286114 |
| Tolleson | 6,989 | 2128242 | NA | 138737 | 19777 | 17065 | 22 | NA |



Table D.5. Manufacturing Data for Municipalities in the Phoenix Active Management Area (Source: US Census Bureau)

| City | Annual payroll ($1,000) | First-quarter payroll ($1,000) | Number of paid employees | Operating expenses ($1,000) | Total inventories, beginning of year ($1,000) | Total inventories, end of year ($1,000) | Sales, receipts, or revenue from administrative records (%) | Sales, receipts, or revenue estimated (%) | Sales Per Establish-ment ($1000) | Payroll Per Establish-ment ($1000) |
|---|---|---|---|---|---|---|---|---|---|---|
| Apache Junction | 1790 | 361 | 66 | 3834 | 3200 | 3582 | 14% | 0% | 1453 | 105 |
| Avondale | NA | NA | NA | NA | NA | NA | 0% | 0% | NA | NA |
| Buckeye | NA | NA | NA | NA | NA | NA | NA | NA | NA | NA |
| Chandler | 291766 | 71408 | 4198 | 543461 | 204970 | 206196 | 1% | 3% | 20845 | 1326 |
| Gilbert | 59745 | 14564 | 1450 | 117011 | 49213 | 50744 | 7% | 13% | 4417 | 406 |
| Glendale | 70030 | 16869 | 2079 | 141645 | 127262 | 133548 | 17% | 5% | 7508 | 519 |
| Goodyear | NA | NA | NA | NA | NA | NA | NA | NA | NA | NA |
| Mesa | 174055 | 35616 | 3372 | 290320 | 479762 | 476581 | 3% | 9% | 6769 | 578 |
| Peoria | 14834 | 4002 | 368 | 31285 | 16012 | 16168 | 6% | 4% | 4486 | 265 |
| Phoenix | 1650697 | 404324 | 34585 | 2989800 | 1655286 | 1669772 | 4% | 7% | 12164 | 848 |
| Scottsdale | 314307 | 75139 | 5811 | 664083 | 334508 | 361302 | 12% | 14% | 6404 | 584 |
| Surprise | 3461 | 862 | 97 | 5980 | 769 | 918 | 5% | 37% | 885 | 150 |
| Tempe | 722174 | 156112 | 11117 | 1201352 | 468771 | 499405 | 4% | 4% | 14259 | 1413 |
| Tolleson | NA | NA | NA | NA | NA | NA | NA | NA | NA | NA |





Table D.6. Retail Data for Municipalities in the Phoenix Active Management Area (source: US Census Bureau)

| City | Population | Number of establishments | Sales ($1,000) | Annual payroll ($1,000) | First-quarter payroll ($1,000) | Number of paid employees for pay period including March 12 | Sales, receipts, or revenue from administrative records (%) | Sales, receipts, or revenue estimated (%) |
|---|---|---|---|---|---|---|---|---|
| Apache Junction | 32,901 | 96 | 447477 | 45386 | 12047 | 2181 | 6% | 8% |
| Avondale | 78,043 | NA | NA | NA | NA | NA | 0% | 0% |
| Buckeye | 37,678 | 48 | 215169 | 14158 | 2830 | 449 | 10% | 26% |
| Chandler | 242,522 | 694 | 3608290 | 353274 | 87134 | 15714 | 2% | 2% |
| Gilbert | 204,904 | 441 | 2079066 | 207222 | 48614 | 8466 | 2% | 4% |
| Glendale | 249,455 | 714 | 3627782 | 332276 | 83526 | 15566 | 3% | 2% |
| Goodyear | 53,654 | 109 | 631710 | 61449 | 16588 | 2955 | 2% | 4% |
| Mesa | 459,742 | 1,507 | 6294523 | 653862 | 164729 | 28855 | 6% | 6% |
| Peoria | 152,795 | 353 | 2340433 | 216892 | 53154 | 8143 | 5% | 2% |
| Phoenix | 1,536,632 | 4,266 | 21859505 | 1913730 | 470361 | 77534 | 7% | 5% |
| Scottsdale | 233,105 | 1,378 | 6645363 | 664928 | 163704 | 22923 | 4% | 5% |
| Surprise | 87,488 | 147 | 888224 | 85661 | 21606 | 4064 | 2% | 1% |
| Tempe | 172,589 | 847 | 6172475 | 447488 | 110713 | 16389 | 4% | 7% |





**Table D.7. Real Estate Data for Municipalities in the Phoenix Active Management Area (source: US Census Bureau)**

| City | Population | Number of establishments | Revenue ($1,000) | Annual payroll ($1,000) | First-quarter payroll ($1,000) | Number of paid employees | Sales, receipts, or revenue from administrative records (%) | Sales, receipts, or revenue estimated (%) |
|---|---|---|---|---|---|---|---|---|
| Apache Junction | 32,901 | 42 | 23469 | 3822 | 1024 | 123 | 12% | 11% |
| Avondale | 78,043 | NA | NA | NA | NA | NA | NA | NA |
| Buckeye | 37,678 | 15 | 46949 | 2305 | 637 | 58 | 2% | 0% |
| Chandler | 242,522 | 301 | 227950 | 37680 | 10260 | 1171 | 13% | 11% |
| Gilbert | 204,904 | 307 | 185181 | 34687 | 8767 | 1003 | 11% | 8% |
| Glendale | 249,455 | 250 | 202866 | 30910 | 7560 | 1208 | 11% | 16% |
| Goodyear | 53,654 | 61 | 33695 | 5577 | 1201 | 166 | 23% | 20% |
| Mesa | 459,742 | 594 | 524907 | 80140 | 20491 | 2834 | 15% | 13% |
| Peoria | 152,795 | 152 | 114499 | 25124 | 6266 | 710 | 11% | 12% |
| Phoenix | 1,536,632 | 2227 | 3261013 | 746350 | 185769 | 17353 | 11% | 12% |
| Scottsdale | 233,105 | 1102 | 1992041 | 335830 | 85047 | 5637 | 9% | 8% |
| Surprise | 87,488 | 55 | 48095 | 6761 | 1653 | 445 | 6% | 6% |
| Tempe | 172,589 | 451 | 768874 | 138486 | 32144 | 3423 | 9% | 9% |
| Tolleson | 6,989 | 3 | 1068 | 370 | 77 | 12 | 39% | 0% |



**Table D.8. Information Services Data for Municipalities in the Phoenix Active Management Area (source: US Census Bureau)**

| City | Population | Number of establishments | Receipts ($1,000) | Annual payroll ($1,000) | First-quarter payroll ($1,000) | Number of paid employees for pay period including March 12 | Sales, receipts, or revenue from administrative records (%)?? | Sales, receipts, or revenue estimated (%)?? |
|---|---|---|---|---|---|---|---|---|
| Apache Junction | 32,901 | 6 | N | 1996 | 492 | 50 | N | N |
| Avondale | 78,043 | NA | N | NA | NA | NA | N | N |
| Buckeye | 37,678 | 5 | N | 933 | 214 | 18 | N | N |
| Chandler | 242,522 | 77 | N | 104599 | 31878 | 2125 | N | N |
| Gilbert | 204,904 | 41 | N | 26622 | 6502 | 454 | N | N |
| Glendale | 249,455 | 40 | N | 22578 | 5956 | 520 | N | N |
| Goodyear | 53,654 | 8 | N | D | D | b | N | N |
| Mesa | 459,742 | 120 | N | 149666 | 39598 | 3006 | N | N |
| Peoria | 152,795 | 26 | N | 9088 | 2390 | 252 | N | N |
| Phoenix | 1,536,632 | 694 | N | 1347304 | 347914 | 21256 | N | N |
| Scottsdale | 233,105 | 249 | N | 541288 | 134269 | 6725 | N | N |
| Surprise | 87,488 | 7 | N | 1593 | 482 | 30 | N | N |
| Tempe | 172,589 | 162 | N | 209756 | 54879 | 4157 | N | N |
| Tolleson | 6,989 | 5 | N | D | D | a | N | N |





Table D.9. Professional and Technical Services Data for Municipalities in the Phoenix Active Management Area (source: US Census Bureau)

| City | Population | Number of establish-ments | Receipts/Revenue ($1,000) | Expenses ($1,000) | Annual payroll ($1,000) | First-quarter payroll ($1,000) | Number of paid employees for pay period including March 12 | Sales, receipts, or revenue from administrative records | Sales, receipts, or revenue estimated (%) | Annual Payroll Per Establish-ment $ |
|---|---|---|---|---|---|---|---|---|---|---|
| Apache Junction | 32901 | 30 | 9679 | N | 2722 | 687 | 122 | 30% | 21% | 91 |
| Avondale | 78043 | NA | NA | NA | NA | NA | NA | NA | NA | NA |
| Buckeye | 37678 | 22 | 22110 | N | 7136 | 1884 | 149 | 2% | 12% | 324 |
| Chandler | 242522 | 610 | 373941 | N | 490947 | 126119 | 7138 | 37% | 18% | 805 |
| Gilbert | 204904 | 463 | 412515 | N | 152162 | 34923 | 3538 | 16% | 6% | 329 |
| Glendale | 249455 | 303 | NA | N | NA | NA | NA | NA | NA | NA |
| Goodyear | 53654 | 67 | 36598 | N | 15146 | 3501 | 355 | 16% | 18% | 226 |
| Mesa | 459742 | 1084 | 709255 | N | 302360 | 69800 | 7120 | 26% | 14% | 279 |
| Peoria | 152795 | 219 | 79141 | N | 27921 | 6860 | 702 | 33% | 8% | 127 |
| Phoenix | 1536632 | 5055 | 7158437 | N | 2947465 | 671593 | 46810 | 15% | 7% | 583 |
| Scottsdale | 233105 | 1972 | 3573147 | N | 1158253 | 272299 | 17313 | 15% | 6% | 587 |
| Surprise | 87488 | 83 | 27484 | N | 11120 | 3024 | 437 | 20% | 22% | 134 |
| Tempe | 172589 | 1005 | 1321432 | N | 606044 | 152654 | 10930 | 15% | 8% | 603 |
| Tolleson | 6989 | 2 | NA | N | NA | NA | NA | NA | NA | NA |



**Data Availability**

The data used in this study is publicly sourced and reproduced in the paper's appendices.

**Author Contributions**

15   BR designed the study and led the writing. RR carried out data collection and calculations and helped with the writing. DH edited and rewrote the manuscript, including preparation of the results.

**Competing interests**

The authors declare that they have no conflict of interest. BR and RR disclose that they were paid consultants to the City of Chandler, Arizona in 2012.

20   **Acknowledgements**

The authors gratefully acknowledge the City of Chandler, Arizona; the City's water resource engineering staff provided advice and data to support the study- in particular Doug Toy and Bob Groff. The authors acknowledge funding from NSF/USDA ACI-1639529, INFEWS/T1: Mesoscale Data Fusion to Map and Model the U.S. Food, Energy, and Water (FEW) System, and from internal funding by Northern Arizona University. This work

25   was conducted as a part of the "Reanalyzing and Predicting U.S. Water Use using Economic History and Forecast Data; an experiment in short-range national hydro-economic data synthesis" Working Group supported by the John Wesley Powell Center for Analysis and Synthesis, funded by the U.S. Geological Survey. The opinions expressed are those of the researchers, and not necessarily the funding agencies.

**References**

30   Arizona Department of Water Resources. 2011a. "Notification of 2009 Gallons per Capita per Day (GPCD) and Lost and Unaccounted (L&U) for Water Percentages. Notification for Arizona Water Company - Apache Junction." Report Number: 56-002000.000. Arizona Department of Water Resources, Phoenix, AZ.

Arizona Department of Water Resources. 2011b. "Second Notification of 2009 Gallons per Capita per Day (GPCD) and Lost and Unaccounted (L&U) for Water Percentages. Notification for City of Avondale."

Report Number: 56-002003.000. Arizona Department of Water Resources, Phoenix, AZ.

Arizona Department of Water Resources. 2011c. "Notification of 2009 Gallons per Capita per Day (GPCD) and Lost and Unaccounted (L&U) for Water Percentages. Notification for Town of Buckeye". Report Number: 56-002006.000. Arizona Department of Water Resources, Phoenix, AZ.



Arizona Department of Water Resources. 2011d. "Notification of 2009 Gallons per Capita per Day (GPCD) and Lost and Unaccounted (L&U) for Water Percentages. Notification for City of Chandler." Report Number: 56-002009.000. Arizona Department of Water Resources, Phoenix, AZ.

Arizona Department of Water Resources. 2011e. "Notification of 2009 Gallons per Capita per Day (GPCD) and Lost and Unaccounted (L&U) for Water Percentages. Notification for Town of Gilbert." Report Number: 56-002017.000. Arizona Department of Water Resources, Phoenix, AZ.

Arizona Department of Water Resources (ADWR), 2011f. "Notification of 2009 Gallons per Capita per Day (GPCD) and Lost and Unaccounted (L&U) for Water Percentages. Notification for City of Glendale." Report Number: 56-002018.000. Arizona Department of Water Resources, Phoenix, AZ.

Arizona Department of Water Resources. 2011g. "Notification of 2009 Gallons per Capita per Day (GPCD) and Lost and Unaccounted (L&U) for Water Percentages. Notification for City of Goodyear." Report Number:
56-002019.000. Arizona Department of Water Resources, Phoenix, AZ.

Arizona Department Of Water Resources. 2011h. "Notification of 2009 Gallons per Capita per Day (GPCD) and Lost and Unaccounted (L&U) for Water Percentages. Notification for City of Mesa." Report Number: 56-002023.00. Arizona Department of Water Resources, Phoenix, AZ.

Arizona Department of Water Resources. 2011i. "Second Notification of 2009 Gallons per Capita per Day (GPCD) and Lost and Unaccounted (L&U) for Water Percentages. Notification for City of Peoria." Report
Number: 56-002029.000). Arizona Department of Water Resources, Phoenix, AZ.

Arnold, C. A. 2005. "Is Wet Growth Smarter than Smart Growth?: The Fragmentation and Integration of Land Use and Water." Environmental Law Reporter 35 (3): 10152–10178. https://www.academia.edu/10850352/The_Most_Important_Current_Research_Questions_in_Urban_Ecosy
stem_Services

Bae, J. and Dall'Erba, S. 2018. "Crop production, export of virtual water and water-saving strategies in Arizona." Ecological Economics 146: 148-156. https://doi.org/10.1016/j.ecolecon.2017.10.018

Bates, S. 2012. "Bridging the Governance Gap: Emerging Strategies to Integrate Water and Land Use Planning." Journal of Natural Resources 52: 61. https://digitalrepository.unm.edu/nrj/vol52/iss1/3

Berg, S. 2010. Water Utility Benchmarking: Measurement, Methodologies and Performance Incentives. London: IWA Publishing.

Blackhurst, B. Y. M., Hendrickson, C., and Vidal, J. S. I. 2010. "Direct and indirect water withdrawals for U.S. industrial sectors." Environmental Science and Technology 44 (6): 2126–2130. https://doi.org/10.1021/es903147k

Borrego-Marín, M. M., Gutiérrez-Martín, C., & Berbel, J. 2016. Estimation of cost recovery ratio for water services based on the system of environmental-economic accounting for water. Water Resources Management, 30 (2): 767-783. DOI:10.1007/s11269-015-1189-2

Central Arizona Project. 2012. "CAP's Subcontracting Status Report for CAP allocations". http://www.cap-az.com/Water/Allocations.aspx, accessed 10 Sep 2012.

City of Chandler. 2009. City of Chandler Annual Budget 2008-09. (http://www.chandlz.gov/content/2008_09AnnualReport.pdf; access date 08/12/2012.

City of Chandler. 2008. Arizona General Plan. (http://www.chandlz.gov/gov/ChandlerGenlPlan.pdf; access date 09/10/2012).

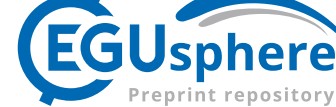

City of Glendale. 2008. Schedule 5: Expenditure Limitation & Property Tax Rate.
(http://www.glendaleaz.com/budget/AnnualBudgetBooks.cfm; access date 8/12/2012).

City of Goodyear. 2007. City of Goodyear 2007-2008 Annual Budget.
(http://www.goodyearaz.gov/DocumentCenter/Home/View/4267; access date 8/12/2012).

City of Mesa. 2008. City of Mesa Final Budget for Fiscal Year Ending 2008.
(http://www.mesaaz.gov/budget/Documents/FY_03_09/Reso_9002_Budget_07_08.pdf; access date
8/12/2012).

City of Peoria, 2007. City of Peoria Annual Program Budget - Fiscal Year 2007.
(http://www.peoriaaz.gov/uploadedFiles/Peoriaaz/Departments/Budget/Historical_Budget_Books/FY2007
AnnualProgramBook.pdf; access date 8/12/2012).

City of Phoenix. 2007. Arizona Comprehensive Annual Financial Report, Financial Year 2007.
(http://phoenix.gov/webcms/groups/internet/@inter/@gov/@fin/@plan/documents/web_content/080342.pd
f: access date 8/12/2012).

City of Scottsdale. 2008. City of Scottsdale Arizona Adopted Financial Year 2007/08 Budget.
(http://www.scottsdaleaz.gov/Assets/Public+Website/finance/Archive/FY+2007-08/FY+2007-
08+Volume+1+Budget+Summary.pdf; access date 8/12/2012.

City of Tempe. 2007. City of Tempe Annual Budget - July 1, 2007 through June 30, 2008.
(http://www.tempe.gov/modules/showdocument.aspx?documentid=631); access date 8/12/2012

Devineni, N., Lall, U., Etienne, E., Shi, D. and Xi, C. 2015. "America's water risk: Current demand and climate
variability." Geophysical Research Letters 42: 2285–2293. https://doi.org/10.1002/2015GL063487

Evenson, E.J., Jones, S.A., Barber, N.L., Barlow, P.M., Blodgett, D.L., Bruce, B.W., Douglas-Mankin, K.,
Farmer, W.H., Fischer, J.M., Hughes, W.B., Kennen, J.G., Kiang, J.E., Maupin, M.A., Reeves, H.W.,
Senay, G.B., Stanton, J.S., Wagner, C.R., and Wilson, J.T. 2018. "Continuing progress toward a national
assessment of water availability and use." U.S. Geological Survey Circular 1440, 64 p.,
https://doi.org/10.3133/cir1440.

Gleick, P. H. and Palaniappan, M. 2010. "Peak water limits to freshwater withdrawal and use." Proceedings of
the National Academy of Sciences of the United States of America 107 (25): 11155–11162.
https://doi.org/10.1073/pnas.1004812107

Gober, P., Kirkwood, C. W., Balling, R. C. Jr., Ellis, A. W. and Deitrick, S. 2010. "Water planning under
climatic uncertainty in Phoenix: Why we need a new paradigm." Annals of the. Association of American.
Geographers. 100 (2): 356– 372. https://doi.org/10.1080/00045601003595420

Gober, P., Larson, K. L., Quay, R., Polsky, C., Chang, H. and Shandas, V. 2013. "Why Land Planners and
Water Managers Don't Talk to One Another and Why They Should!" Society and Natural Resources 26 (3):
356–364. https://doi:10.1080/08941920.2012.713448

Haider, H., Sadiq, R., and Tesfamariam, S. 2016. Inter-Utility Performance Benchmarking Model for Small-to-
Medium-Sized Water Utilities: Aggregated Performance Indices. Journal of Water Resources Planning and
Management 142(1). https://ascelibrary.org/doi/full/10.1061/%28ASCE%29WR.1943-5452.0000552

Hamdy, A., Ragab, R. and Scarascia-Mugnozza, E., 2003. Coping with water scarcity: water saving and
increasing water productivity. Irrigation and Drainage: The Journal of the International Commission on
Irrigation and Drainage, 52(1), pp.3-20.



Hoekstra, A. Y., Chapagain, A. K., Aldaya, M. M. and Mekonnen, M. M. 2011. "The water footprint assessment
   manual: Setting the global standard." London: Earthscan.

Hoekstra, A. Y. 2014. "Sustainable, efficient, and equitable water use: The three pillars under wise freshwater
   allocation." Wiley Interdisciplinary Reviews Water 1 (1): 31–40. https://doi.org/10.1002/wat2.1000

Hoekstra, A. Y., Chapagain, A. K. and Zhang, G. 2015. "Water footprints and sustainable water allocation."
   Sustainability 8 (1): 20. https://doi.org/10.3390/su8010020

Holway, J. M. 2007. Urban growth and water supply. Arizona Water Policy: Management Innovations in an
   Urbanizing, Arid Region, 157-172.

Howe, C. W., and Goemans, C. 2007. "The simple analytics of demand hardening." Journal-American Water
   Works Association 99 (10): 24-25. https://doi.org/10.1002/j.1551-8833.2007.tb08052.x

Hubler, D. K., Baygents, J. C., Mackay, C., Megdal, S. B. 2012. "Evaluating economic effects of semiconductor
   manufacturing in water-limited regions." Journal of the American Water Works Association 104:2.
   https://doi:10.5942/jawwa.2012.104.0024

Jacobs, K., & Megdal, S. 2004. Water management in the active management areas. Arizona's Water Future:
   Challenges and Opportunities Background Report, 71-94.

Kijne, J.W., Barker, R. and Molden, D., 2003. Improving water productivity in agriculture: editors' overview.
   Water productivity in agriculture: Limits and opportunities for improvement, pp.xi-xix.

Larson, K. L., Polsky, C., Gober, P., Chang, H. and Shandas, V. 2013. "Vulnerability of Water Systems to the
   Effects of Climate Change and Urbanization: A Comparison of Phoenix, Arizona and Portland, Oregon
   (USA)." Environmental Management 52: 179–195. https://doi.org/10.1007/s00267-013-0072-2.

Li, E., Li, S. and Endter-Wada, J. 2016. "Water-smart growth planning: linking water and land in the arid
   urbanizing American West." Journal of Environmental Planning and Management 60 (6): 1056-1072,
   https://doi.org/10.1080/09640568.2016.1197106

Maricopa County Department of Finance. 2007. Maricopa County 2007 Tax Levy.
   http://www.maricopa.gov/Finance/PDF/Tax/TaxLevy2007.pdf; access date 08/11 2012).

Marston, L. and Cai, X. 2016. "An overview of water reallocation and the barriers to its implementation." Wiley
   Interdisciplinary Reviews Water 3 (5): 658–677. https://doi.org/10.1002/wat2.1159

Marston, L., Ao, Y., Konar, M., Mekonnen, M. M. and Hoekstra, A. Y. 2018. "High-resolution water footprints
   of production of the United States." Water Resources Research 54: 2288–2316.
   https://doi.org/10.1002/2017WR021923.

Marston, L.T., Lamsal, G., Ancona, Z.H., Caldwell, P., Richter, B.D., Ruddell, B.L., Rushforth, R.R. and Davis,
   K.F., 2020. Reducing water scarcity by improving water productivity in the United States. Environmental
   Research Letters, 15(9), p.094033.

Maupin, M. A., Kenny, J. F., Hutson, S. S., Lovelace, J. K., Barber, N. L. and Linsey, K. S. 2014. "Estimated
   Use of Water in the United States in 2010." Circular 1405. Reston, VA: U.S. Geological Survey.
   http://pubs.usgs.gov/circ/1405/.

Mayer, A., Mubako, S. and Ruddell, B.L., 2016. Developing the greatest Blue Economy: Water productivity,
   fresh water depletion, and virtual water trade in the Great Lakes basin. Earth's Future, 4(6), pp.282-297.



Paterson, W. Rushforth, R., Ruddell, B.L., Ikechukwu, C., Gironás, J., Konar, M., Mijic, A., Mejia, A. 2015. "Water Footprint of Cities: A Review and Suggestions for Future Research." Sustainability 7: 8461-8490. https://doi.org/10.3390/su7078461

Postel, S. L., Daily, G. C. and Ehrlich, P. R. 1996. "Human appropriation of renewable fresh water." Science 271 (5250): 785–787. https://doi.org/10.1126/science.271.5250.785

Richter, Brian D., et al. "Decoupling Urban Water Use and Growth in Response to Water Scarcity." Water 12.10 (2020): 2868.

Ruddell, B. L. (2018) HESS Opinions: How should a future water census address consumptive use? (And where
can we substitute withdrawal data while we wait?), Hydrol. Earth Syst. Sci., 22, 5551–5558, https://doi.org/10.5194/hess-22-5551-2018.

Rushforth, R. R., Adams, E. A. and Ruddell, B. L. 2013. "Generalizing ecological, water and carbon footprint methods and their worldview assumptions using Embedded Resource Accounting". Water Resources and Industry, 1: 77-90. https://doi.org/10.1016/j.wri.2013.05.001

Ruddell, B. L., Adams, E. A., Rushforth, R. and Tidwell, V. C. 2014. "Embedded resource accounting for coupled natural-human systems: An application to water resource impacts of the western US electrical energy trade." Water Resources Research 50: 7957– 7972. https://doi.org/10.1002/2013WR014531

Rushforth, R. R. and Ruddell, B. L. 2015. "The hydro-economic interdependency of cities: Virtual water connections of the Phoenix, Arizona metropolitan area." Sustainability 7 (7): 8522–8547.
https://doi.org/10.3390/su7078522

Rushforth, R. R. and Ruddell, B. L. 2016. "The vulnerability and resilience of a city's water footprint: The case of Flagstaff, Arizona, USA." Water Resources Research 52: 2698–2714. https://doi.org/10.1002/2015WR018006S

Rushforth, R. R. and Ruddell, B. L. 2018. "A spatially detailed blue water footprint of the United States
economy." Hydrology and Earth System Sciences 22: 3007–3032, https://doi.org/10.5194/hess-22-3007-2018.

Rushforth, Richard R., Maggie Messerschmidt, and Benjamin L. Ruddell. "A Systems Approach to Municipal Water Portfolio Security: A Case Study of the Phoenix Metropolitan Area." Water 12.6 (2020): 1663.

Schewe, J., Heinke, J., Gerten, D., Haddeland, I., Arnell, N. W., Clark, D. B., Dankers, R., Eisner, S., Fekete,
B.M., Colón-González, F.J., Gosling, S.N., Kim, H., Liu, X, Masaki, Y, Portmann, F.T., Satoh, Y., Stacke, T., Tang, Q., Wada, Y., Wisser, D., Albrecht, T., Frieler, K., Piontek, F., Warszawski, L., Kabat, P. 2014. "Multimodel assessment of water scarcity under climate change." Proceedings of the National Academy of Sciences 111 (9): 3245–3250. https://doi.org/10.1073/pnas.1222460110S

Scott, C.A. and Pasqualetti, M.J. 2010. "Energy and water resources scarcity: Critical infrastructure for growth
and economic development in Arizona and Sonora." Natural Resources Journal 50 (3): 645-682. JSTOR, www.jstor.org/stable/24889651

Solley, W. B., Chase, E. B. and Mann IV, W.B. 1983. Estimated use of water in the United States in 1980. USGS Circular 1001, U.S. Geological Survey, Washington D.C. https://doi.org/10.3133/cir1001

Tidwell, V. C., Kobos, P. H., Malczynski, L. A., Klise, G. and Castillo, C. R. 2012. "Exploring the water-
thermoelectric power nexus.", Journal of Water Resources Research, Planning and Management 138 (5): 491– 501. https://ascelibrary.org/doi/abs/10.1061/%28ASCE%29WR.1943-5452.0000222



Town of Avondale. 2010. Annual Budget & Financial Plan - Fiscal Year 2010-2011. (http://www.avondale.org/documents/22/54/56/Avondale%20Budget%20Document%20INet.pdf; access date 8/12/2012).

Town of Buckeye. 2007. Arizona Adopted Budget Fiscal Year 2007/08. (http://www.buckeyeaz.gov/DocumentCenter/Home/View/490; access date 8/12/2012).

Town of Gilbert, 2008. Summary Schedule of Estimated Revenues and Expenditures/Expenses. Fiscal Year 2007-08.

(http://www.gilbertaz.gov/budget/pdf/schedule/FY08%20Gilbert%20Official%20C&T%20Budget%20Sche
dule%20A.pdf; access date 8/12/2012).

U.S. Bureau of Economic Analysis. 2019. https://www.bea.gov/data/gdp/gdp-county-metro-and-other-areas

U.S. Census Bureau. 2007a. "American Community Survey, Information: Geographic Area Series: Summary Statistics for the United States, States, Metro and Micro Areas, Metro Divisions, Consolidated Cities, Counties, and Places: 2007." (http://factfinder2.census.gov; access date 08/11/ 2012). Now: "City and
210 Town Intercensal Data sets: 2000 – 2010" (https://www.census.gov/data/datasets/time-series/demo/popest/intercensal-2000-2010-cities-and-towns.html).

U.S. Census Bureau. 2007b. "American Community Survey, All sectors: Geographic Area Series: Economy-Wide Key Statistics". (2007http://factfinder2.census.gov; access date 08/11/ 2012). Now 'Economic Census (2017, 2012, 2007, 2002)." (https://www.census.gov/data/developers/data-sets/economic-
215 census.2007.html).

U.S. Census Bureau. 2009a. "American Community Survey, Selected Economic Characteristics: 2005-2009." (http://factfinder2.census.gov; access date 08/11/ 2012).

U.S. Census Bureau. 2009b. "American Community Survey, Selected Housing Characteristics: 2005-2009." (http://factfinder2.census.gov; access date 08/11/ 2012).

U.S. Census Bureau. 2009c. "American Community Survey, ACS Demographic and Housing Estimates: 2005-2009." (http://factfinder2.census.gov; 08/11/2012).

U.S. Census Bureau. 2009d. "American Community Survey, Mean Income in the Past 12 Months (In 2009 Inflation-Adjusted Dollars)." (http://factfinder2.census.gov; access date 08/11/2012).

U.S. Census Bureau. 2009e. "American Community Survey, Median Income in the Past 12 Months (In 2009
Inflation-Adjusted Dollars)." (http://factfinder2.census.gov; access date 08/11/2012).

U.S. Census Bureau. 2009f. "American Community Survey, Financial Characteristics." (http://factfinder2.census.gov; access date 08/11/ 2012).

U.S. Census Bureau. 2020. https://www.census.gov/newsroom/press-releases/2020/pop-estimates-county-metro.html

U.S. Census (2021). QuickFacts. URL: https://www.census.gov/quickfacts/fact/table/US/PST045219

Vardon, M., Martinez-Lagunes, R., Gan, H., & Nagy, M. 2012. The system of environmental-economic accounting for water: development, implementation and use. Water Accounting, International Approaches to Policy and Decision Making. Edward Elgar, United Kingdom, 32-57. https://DOI: 10.4337/9781849807494.00010

Vörösmarty, C. J. 2000. "Global water resources: Vulnerability from climate change and population growth." Science 289 (5477): 284-288. https://doi.org/10.1126/science.289.5477.284



Wildman, R. A., Jr., and N. A. Forde. 2012. "Management of water shortage in the Colorado river basin: Evaluating current policy and the viability of interstate water trading." Journal of the American Water Resources Association 48 (3): 411– 422. https://doi.org/10.1111/j.1752-1688.2012.00665.x

Xu, Z., Chen, X., Wu, S.R., Gong, M., Du, Y., Wang, J., Li, Y. and Liu, J., 2019. Spatial-temporal assessment of water footprint, water scarcity and crop water productivity in a major crop production region. Journal of Cleaner Production, 224, pp.375-383.



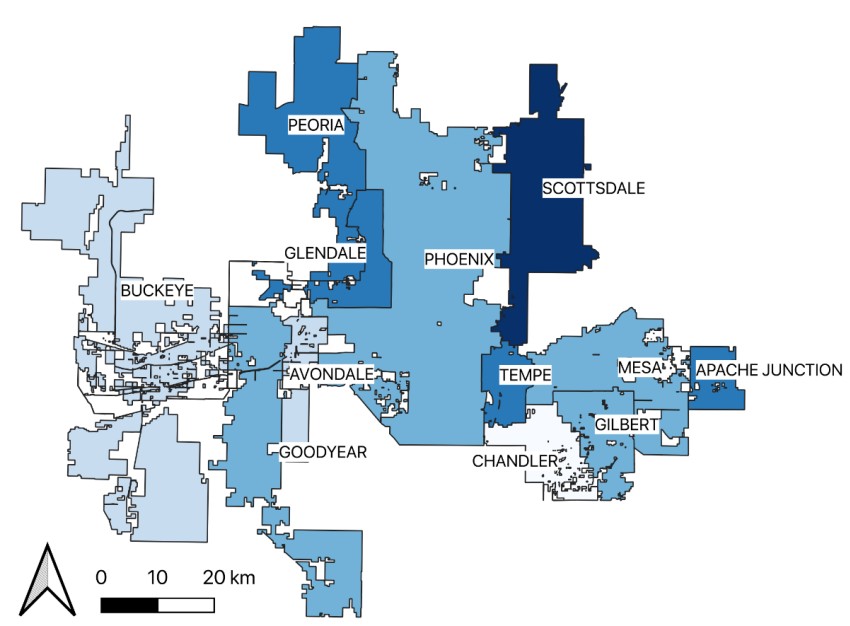

**Figure 1. Map of the Phoenix metropolitan statistical area (PMA) showing the member municipalities.**



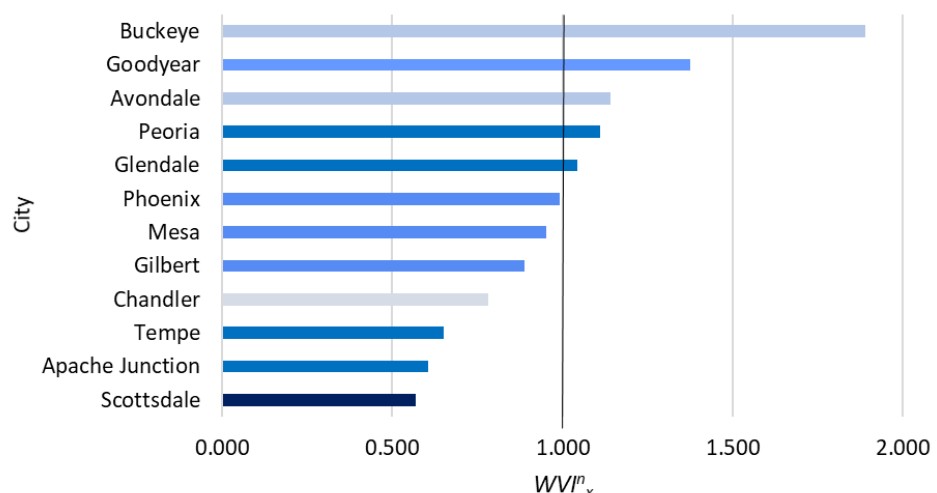

**Figure 2. PMA municipalities x listed in order of their relative WVI$^n_x$ for residential population supported. The PMA's mean value is 1. Outlying bedroom communities like Buckeye, Goodyear, and Avondale score above average on the traditional per-capita basis of water use benchmarking (cities are color-coded to correspond with Figure 1).**



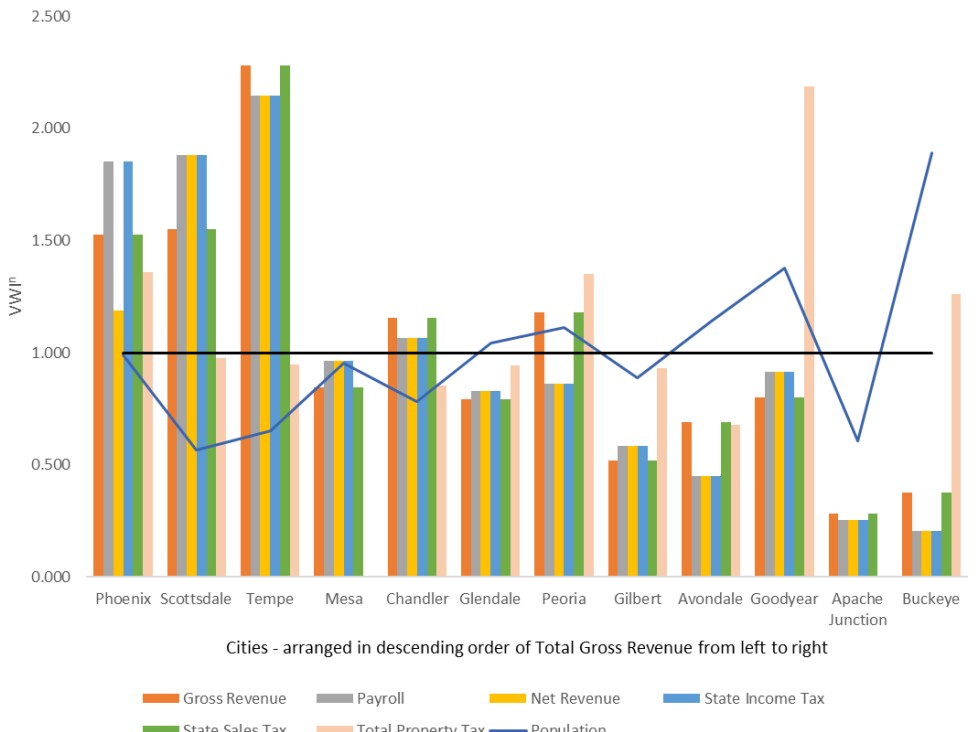

**Figure 3.** *WVIⁿ* **for economic value types (colored bars) and population value type (blue line) for each PMA municipality. The PMA's mean value is 1 (black). Municipalities are arranged in order of decreasing tax revenues from left to right. This ranking also corresponds approximately with geographic distance from the overall urban center of Phoenix, and to size of population and economic GDP. Core municipalities like Tempe, Scottsdale, and Phoenix score above average on an economic basis of water use benchmarking, but below average on a population basis of population supported, demonstrating some degree of tradeoff between these productivity objectives.**





Table 1. General Characteristics of Cities in the Phoenix Metropolitan Statistical Area (*no reclaimed water)

| City | Population | Area (km²) | Density (pop km⁻²) | Payroll ($x1000) | Gross Revenue ($x1000) | Total Property Tax ($x1000) | Income Tax ($x1000) | Sales Tax ($x1000) | Total Water Use* (ac-ft) | Acre Feet Per Km² of City | Acre Feet Per Person Per Km² |
|---|---|---|---|---|---|---|---|---|---|---|---|
| Apache Junction | 32,901 | 91 | 362 | 3,364 | 42,344 | NA | 108 | 2,795 | 10,244 | 759 | 28 |
| Avondale | 78,043 | 119 | 656 | 7,534 | 129,608 | 5,883 | 241 | 8,554 | 12,119 | 689 | 18 |
| Buckeye | 37,678 | 971 | 39 | 990 | 20,512 | 3,186 | 32 | 1,354 | 4,989 | 34 | 129 |
| Chandler | 242,522 | 166 | 1460 | 80,685 | 987,115 | 33,616 | 2,582 | 64,150 | 23,501 | 945 | 16 |
| Gilbert | 204,904 | 166 | 1234 | 32,876 | 330,022 | 22,258 | 1,052 | 21,782 | 44,335 | 1,782 | 36 |
| Glendale | 249,455 | 155 | 1609 | 48,376 | 521,636 | 28,557 | 1,548 | 34,428 | 69,359 | 2,994 | 44 |
| Goodyear | 53,654 | 495 | 108 | 8,702 | 85,775 | 10,805 | 278 | 5,661 | 11,169 | 150 | 104 |
| Mesa | 459,742 | 352 | 1306 | 133,398 | 1,121,299 | NA | 3,628 | 74,006 | 105,459 | 2002 | 80 |
| Peoria | 152,795 | 451 | 339 | 28,945 | 445,973 | 23,529 | 926 | 29,434 | 51,437 | 764 | 153 |
| Phoenix | 1,536,632 | 1339 | 1148 | 700,624 | 6,504,679 | 266,891 | 22,420 | 429,809 | 377,341 | 1,891 | 329 |
| Scottsdale | 233,105 | 477 | 489 | 188,927 | 1,750,749 | 50,838 | 6,046 | 115,549 | 109,065 | 1,536 | 223 |
| Tempe | 172,589 | 104 | 1660 | 138,748 | 1,658,540 | 31,736 | 4,439 | 109,463 | 70,907 | 4,600 | 41 |