# Peer review of "Water productivity is in the eye of the beholder: benchmarking the multiple values produced by water use in the Phoenix metropolitan area"

_EGUsphere, 2022_

## Author Response (AR1)

**Reply to van der Zaag; replies inline**

I find the topic interesting and relevant. I have four comments and a few more detailed comments.

My first comment is that I find the paper not so easy to read, although the argument of the paper is not very complex. I am therefore convinced that the presentation of the work can be much improved. I mention two points of improvements.

(a) The authors could select a reduced number of water productivity metrics that most aptly reflect the point the authors wish to make (and in supplementary materials they could include details of the other metrics considered). I base this comment on the following. Figure 3 demonstrates that two groups of each two metrics behave identically in the different locations (namely Gross Revenue and State Sales Tax, and Payroll and State Income Tax), and for the second group a third metric only deviates from its two metrics in one location (namely Net Revenue, which behaves differently only in Phoenix). If one would delve into the details of those metrics, I guess it wouldn't be very difficult to explain why they behave identically. This would enable to simplify the paper and select the most telling metric of each group, or the one that is most easily to be collected/measured, while commenting that the other metric can also be used.

- AUTHORS: It is true that these metrics are correlated, particularly with respect to taxes which are a simple function of profits. But they are not correlated 1:1. Because we have not seen a similar paper published in the literature in the past, one of our goals is to present as many different value metrics as possible, and to give the reader a basic sense of how they are related. We strongly agree that a real-world policy application of this method would choose a subset of metrics (and possibly different metrics than are presented here). The argument about "point of view" is the important idea here. Different political constituencies will care about different values, so it is important to calculate and present the values that each constituency cares about, even if those values are redundant or correlated. This is actually desirable that multiple constituencies have correlated values, as it aligns interests and (often) policy preferences.

(b) The point made towards the end of section 2 could be presented more prominently, namely that different stakeholders value different metrics differently. The argument of the paper could then be weaved along these selected stakeholders. This would in my view make the paper much more easily readable and thus much more attractive.

- AUTHORS: Thank you for emphasizing this point. We agree, and we have promoted this argument. This argument is really central to the purpose of this study, and it is important that the reader not be allowed to miss the point.

As a side note, it remains unclear how many metrics the paper reports; in the text it is stated that they are six (line116), whereas 7 are reported in lines 116 and 117, but in lines 80-81, there are 8 metrics mentioned. But note that "net revenue" is mentioned twice in both. In figure 3, however, only 6 appear. This requires clarification.

- AUTHORS: Agreed – we have clarified the text so that there is no mention of a specific number of metrics to avoid confusion throughout the text.

My second comment concerns details. Some details are given in the paper, but these should either be better explained, or relegated to an appendix. I refer here to the issue of including or excluding the indirect value. I guess this is an important issue but the assumptions and implications of excluding these seem to come as an afterthought.

- AUTHORS: The second reviewer also brought up the issue of indirect value. Indirect value presents us with a difficult editorial decision. On one hand, indirect value creation is a real thing and is relevant to economic decisions; and it is also a part of the Embedded Resource Accounting framework we apply in the paper's mathematics. On the other hand, indirect value creation is secondary to the direct value creation, and including those results will make the paper dramatically longer and more complicated. We decided to explicitly recognize indirect value creation as a component of the mathematics, and then to neglect it for clarity and brevity, as it is secondary to our main point. It might be clearer to simply ignore the topic completely, as most related papers choose to do, but in our view this fails the reader. We have attempted to clarify the implications of excluding indirect value, and we hope you find the effort an improvement. References:

  Ruddell, B.L., Adams, E.A., Rushforth, R. and Tidwell, V.C., 2014. Embedded resource accounting for coupled natural-human systems: An application to water resource impacts of the western US electrical energy trade. Water Resources Research, 50(10), pp.7957-7972.

  Rushforth, R.R. and Ruddell, B.L., 2015. The hydro-economic interdependency of cities: Virtual water connections of the Phoenix, Arizona Metropolitan Area. Sustainability, 7(7), pp.8522-8547.

A third comment: I am not an economist, but I find the suggestion that all values used as metrics are all fully attributed to (or imputed to) the water input. For example, in line 202 it states: "Property taxes were used as a measure for the values produced by residential water use." Similarly, in lines 94-95 it is stated that "Within each municipality water is allocated to Residential and Non-Residential uses, which yield residential values (income tax, property tax, population) and non-residential values (payroll, net/gross revenue, sales tax)."

But it is clear that water is only one input among many that produces this value, and often a relatively minor input compared to others.

- AUTHORS: We do not intend to argue that all the values used as metrics are fully attributed to the water. That would be the "Value of the Marginal Product" of the water (under SEEA rules). Instead, we are presenting numbers more like Water Productivity, as you point out; it is the simple ratio of a partial output to a partial input.

A fourth comment refers to the statement made in the beginning of the paper in lines 49-50: "... we manage what we measure ...". This is potentially a problematic statement, as this might imply that what cannot be measured cannot be managed. And this may have a problematic relation with a remark made towards the end of the paper, namely in lines 278-280, "We omit environmental and social wellness values, from this research (…) due to a lack of quantifiable data on these measures." Doesn't this pose a fundamental limitation to the main purpose of this paper, namely that certain important values may not be quantifiable in water productivity metrics. I would find it interesting if the authors would include a more nuanced reflection on this issue.

- AUTHORS: We agree that the lack of environmental and social data (and other economic data) is a limitation of the study. The paper would be better if we had more of this data. Hopefully future applications of this approach will solve that limitation. We do not wish to scientifically argue the statement that "we manage what we measure", as it is a figure of speech meant to add emphasis. In fact, Deming agrees with you that the statement is not entirely correct. We agree to remove the statement if it does more harm than good. Therefore, we have removed the Deming sentence and qualified our study findings with the following statement:

  "Because there are many social, environmental, and economic stakeholders with many different sets of interests and values, multiple water use efficiency or productivity benchmarks are appropriate to measure the efficacy of water allocation. Although it should be noted that current study did not include the social, environmental and full economic value of water due to a lack of available data."

More detailed comments:

The title is quite general – perhaps a subtitle could indicate what specifically the paper is about, namely that water adds different types of value that cannot easily be commensurated.

- AUTHORS: We have changed the title to: "Water productivity is in the eye of the beholder; the multiple values produced by water use in a metropolitan area"

I would prefer a straightforward use of the concept of water productivity, as against the little known "water value intensity" concept.

- AUTHORS: We have made straightforward comparison/linkage to water productivity in the following passage:

  "Per Kumar (2021), we present here as WVI is similar to the water productivity definition based on single factor of production using water use. In other words, WVI is similar to the Partial factor productivity (PFP), which is a ratio of a measure of total output to a measure of a single input category. The two differences are technicalities, and are that (a) WVI could include indirect value production, and (b) WVI makes no attempt to use total productivity and instead is calculated several times using several different and non-commensurable productivities (i.e. values). WVI is a specific and disambiguated metric that is roughly similar to water productivity but differs in the precise details of its construction. However, WVI is a very specific type of water productivity measurement that facilitates comparison of non-commensurable values."

The values of WVI presented (as in Figure 3) are, I guess, normalized values. But this is not explained in the text. Also the dimensions/units of value should be explicitly reported when values are presented.

- AUTHORS: Addressed.

The concept of "payroll' may need an explanation, as not all readers may be familiar with this concept; I understand that it refers to the wage bill/salaries?

- AUTHORS: Agreed, and you are correct with your understanding. We have addressed this in Line 57.

The acronyms SRP and CAP are not explained when first used (line 92), nor ADWR (line 108).

- AUTHORS: Addressed.

I found it confusing that one the one hand the authors explain that they use data on water withdrawals, and not data on water consumed (line 102 and lines 107-112), and that they later add L&U water (= lost and unaccounted for water) to the data on water use/water withdrawals they got from ADWR. To the reader this may seem that some double counting of water might occur. This requires an explanation.

- AUTHORS: Thank you for pointing out the confusion. The difference here is not between withdrawal and consumption, but rather between withdrawn and delivered water. Delivered water is withdrawn water less L&U losses. Delivered water is the right way to count from the perspective of the water customer and is used herein, but withdrawn water could be the right way to count from the city's perspective.

In this same sense I think it would be wise to use one consistent term for these data: is it "water use", "water withdrawal" or "water demand"? In the tables in Annex B the term 'demand' is used. But the demand for water may not be identical to the use of water, and thus not for the water withdrawals. So water demand is not entirely a correct term for the data used in this paper. I oudl recommend to choose between "water use", "water withdrawal".

- AUTHORS: Agreed. We have simplified the language to "water use" and defined it as water delivered less L&U water.

Appendix A-1 (line 140) doesn't exist. I guess you mean Appendix A.

- AUTHORS: Agreed. Fixed.

Lines 156-157 state: "In this case there are six direct and local values produced, one direct impact on the local freshwater stock…" But these direct and local values and the direct impact are not made explicit. Are these the same as the (6, or 67 or 8) mentioned earlier?

- AUTHORS: Agreed. Sentence clarified to read:

  "In this case there are direct and local values produced (e.g., Tables D1-D9), direct impact on the local freshwater stock, and indirect values and impacts are neglected."

Figure 3: not clear what the blue line of population signifies. What is the unit? What does a population of "1" mean?

- AUTHORS: Addressed; Clarified unit as population per acre-foot on Figure 3.

Table 1: I don't understand how the last two columns ("Acre feet per km2 of city" and "Acre feet per person per km2") were calculated.

AUTHORS: This explanation was not provided; we will add it.

There are several sentences where a word is missing, or something else is amiss in the sentence, including in the following lines: 99, 109, 118, 129, 266, 280.

AUTHORS: ADDRESSED.

Citation: https://doi.org/10.5194/egusphere-2022-1367-AC1

This manuscript evaluates water use volumes in Phoenix metropolitan area cities by generating different metrics that are based on population, taxes, revenues etc. The aim is to demonstrate to decisionmakers that the conventional water allocation metrics, water use volume per inhabitant, is insufficient and may result in biased or misleading policies regarding the allocation of (scarce) water resources.

As such I believe the concept of the paper is of value for different stakeholders in the US (and beyond) and help to understand that a broader set of metrics should be used when shaping water allocation policies. For the scientific audience, the manuscript may be of less interest as I believe a majority of the findings are somewhat self-evident. For instance, that the pattern of water use (withdrawal) is dependent on the relative prevalence of (the different types of) primary production, manufacturing industries, services and residential areas. Therefore, to make the manuscript more interesting for larger scientific audiences, I make some suggestions how to elaborate the present version. Also, I spotted some minor issues that the authors should check and correct.

Major comments:

In the introduction, the authors correctly state that the simple metrics, Gallons per Capita per Day (GPCD), simplifies water allocation policies unnecessarily, if used as such. The authors, however, also state that "[t]he goal of water policy should be to do more social, environmental, and economic good with limited water resources, but not necessarily to use less water." While this may be true, this phrasing made me ask whether water scarcity is an issue in the case study area (as suggested in the Introduction)? For the choice of water allocation metrics and policies I suppose it would make a lot of difference if there is a need to limit the overall water withdrawal within the case study area. I might have missed this notion and connection in the manuscript but for me it remained unclear which role the simple need to reduce water use really plays in the study area. This should be clarified in the manuscript. Are you expecting to run short of available water and is that the primary incentive for the need of different metrics? I wonder if you could clarify this.

- AUTHORS: Thank you for pointing out the lack of clarity. We see that we assumed readers know about the serious water stresses and shortages facing the Phoenix Metropolitan Area. We will add clarity on this point. We have addressed this comment by modifying the statement to read:

  "The goal of water policy should be to do more social, environmental, and economic good with limited water resources, but not necessarily to use less water but to maximize the value of scarce resource, which may include conservation measure that allow for the future use of water."

Following that thought, it would be essential to evaluate what kind of water use (industries, activities) are promoted in political decision-making. Is it possible to increase the share of water-intensive industries or the other way around? How about the anticipated (?) increase in the population?  This is also related to the fact that you analyzed, as far as I understood it right, only direct water use and embedded or life-cycle water use is not addressed. Bringing these two issues together, decision-makers

might want to prioritize water allocation to industries/activities that is essential for the local supply chains and residents. For instance, if there is food industry that is dependent on local agricultural commodities, it would be valuable to identify such value chains and analyze their role in the water allocation and related policies. Similarly, if for instance semiconductor industry would rely largely on resources from other regions and countries, such interdependencies would not be related to water use in regard to that industry. I realize that you might not have data available to address these kinds of issues. Nevertheless, I wonder if these issues would find a place in the Discussion in the context of the use and need of different water use metrics.

- AUTHORS: Your question about indirect water usage in the value chain is apt and was also raised by an earlier reviewer; please see our answer to the other reviewer on the point. In fact the authors have already published the analysis you mention here- specifically, on the virtual water in supply chains that lie within and without the Phoenix Metropolitan Area. Combining that indirect water use analysis with the present paper's multiple value analysis is a very good idea, but it is outside our current scope. We think it is important to develop a clear presentation of the multiple-values argument first, and on its own merits, before adding the complication of indirect valuation. We have added the following passage to the end of the second to last paragraph of Section 4:

  "Additionally, combining indirect water use analysis (Rushforth and Ruddell, 2015) with the present paper's multiple value analysis to provide a complete evaluation of the value crated by water use, but it is outside the scope of this work. We think it is important to develop a clear presentation of the multiple-values argument first, and on its own merits, before adding the complexity of indirect value creation from water use."

Minor comments:

Graphical abstract: The meaning of the black line is not explained. The caption of Fig. 3 states that "the PMA's mean value is 1 (black)". Please consider adding this text to the legend box.

AUTHORS: The black line indicates a value intensity of 1.

1. 82: Net revenue is mentioned two times, in (3) and (5).

   - AUTHORS: Addressed.

2. 93: The abbreviations SRP and CAP are explained only at line 163. They should be explained here.

   - AUTHORS: Addressed.

3. 111: "Also, reclaimed water generally is used low economic value or indirect economic value… " Is there something missing here?

- AUTHORS: Rephrased to: "Also, we do not consider the indirect value of reclaimed water because the reclaimed water uses, such as recreational turf irrigation, make it difficult to measure associated economic value. Additionally, reclaimed water (unlike potable water) is subject to varied city and county policies and standards for reporting and accounting, making it is difficult to compare reclaimed water data robustly between municipalities."

4. 149: "Simplified Embedded Resource Accounting: or, Point of View Matters in Water Use Accounting" Are these subtitles alternative to each other?

- AUTHORS: Addressed.

5. 156 "is therefore also disinterested in in indirect value creation…" Extra 'in' in this sentence?

- AUTHORS: Addressed.

6. 168-183. This part of the method description is difficult to comprehend. Also, later in the manuscript, there is a reference to 'Pareto'. I believe this text is easier to digest if one is an economist but for larger audiences this might be too ambiguous.

- AUTHOR: Simplified the text to read:

  "*WVI*'s may include economic data and measures of economic value, but a *WVI* – or any *VI* – is not a price or a measure of marginal value or cost according to the classical economic *Theory of Value*, because it does not consider the marginal contribution of the impact on the resource stock to the production of values, or the cost of the resource, or value-added by the process. Since *VI*'s are not prices or costs, they may not be added together to directly measure the value produced by a process. Rather, *VI*'s should be interpreted as multiple independent objectives or assigned relative weights by a decision maker."

AUTHORS: Agreed; we will address these points.
Citation: https://doi.org/10.5194/egusphere-2022-1367-AC2